# ABA Enhances Drought Resistance During Rapeseed (*Brassica napus* L.) Seed Germination Through the Gene Regulatory Network Mediated by ABA Insensitive 5

**DOI:** 10.3390/plants14091276

**Published:** 2025-04-22

**Authors:** Dan Luo, Qian Huang, Manyi Chen, Haibo Li, Guangyuan Lu, Huimin Feng, Yan Lv

**Affiliations:** 1Guangdong Provincial Key Laboratory for Green Agricultural Production and Intelligent Equipment, Guangdong University of Petrochemical Technology, Maoming 525000, China; 2Oil Crop Research Institute, Chinese Academy of Agricultural Science, Wuhan 430062, China; 3Institute of Crop Science, Zhejiang University, Hangzhou 310058, China; 4Henry Fok College of Biology and Agriculture, Shaoguan University, Shaoguan 512005, China

**Keywords:** *Brassica napus* L., *ABI5*, CRISPR-CAS9, transcriptome, drought tolerance, seed germination

## Abstract

ABA Insensitive 5 (ABI5) is a basic leucine zipper (bZIP) transcription factor (TF) that plays a critical role in seed dormancy and germination, particularly under stress conditions. This study identified *ABI5* as an important candidate gene regulating seed germination under drought stress during early germination in rapeseed (*Brassica napus* L.) seeds through Genome-Wide Association Study (GWAS). Using Clustered Regularly Interspaced Short Palindromic Repeats/CRISPR-associated protein 9 (CRISPR/CAS9) technology, *ABI5* mutant plants were generated, showing higher germination rates and more developed root systems at 72 h. Transcriptomic analysis of wild-type (WT) and mutant seeds under water, 2μM of abscisic acid (ABA), and 10% PEG treatments after 0, 24, 48, and 72 h revealed complex changes in gene regulatory networks due to *ABI5* mutation. Differential expression analysis showed that the number of downregulated differentially expressed genes (DEGs) in the mutant was significantly higher than upregulated DEGs at multiple time points and treatments, indicating a negative regulatory role for *ABI5* in gene expression. Weighted Gene Co-Expression Network Analysis (WGCNA) revealed that genes related to ABA content, such as those in the glutathione metabolism pathway, were similarly downregulated in the *ABI5* mutants. Key genes, including *BnA03g0120550.1* (*GST*), *BnA09g0366300.1* (*GST*), *BnA10g0413960.1* (*gshA*), and *BnC02g0518750.1* (*GST*), were identified as potential candidates in *ABI5*-regulated drought responses. Additionally, TFs involved in regulating the glutathione metabolism pathway were identified, providing insights into the collaboration of *ABI5* with other TF. This comprehensive transcriptomic analysis of *ABI5* mutant plants highlights how *ABI5* affects gene expression in multiple pathways, impacting seed germination and drought resistance, offering a foundation for improving drought tolerance in rapeseed.

## 1. Introduction

Drought stress is one of the most common abiotic stresses affecting global agricultural production, particularly in regions with limited water resources [1]. Over the course of evolution, plants have developed various mechanisms to cope with drought, including morphological and biochemical adaptations such as changes in leaf morphology, thickening of the waxy cuticle, and osmoregulation [2,3,4]. Additionally, plant hormones, including jasmonic acid, ethylene, and abscisic acid (ABA), play crucial regulatory roles in drought responses [5,6].

The relative levels of plant hormones regulate seed dormancy and germination. Gibberellins (GA), brassinosteroids, ethylene, and cytokinins have been shown to break seed dormancy and promote germination [7,8,9]. In contrast, ABA is the only known hormone capable of inducing and maintaining seed dormancy [10], and it plays a key role in regulating drought tolerance, seed dormancy, and germination [11,12]. Other plant hormones, such as GA, ethylene, cytokinins, and brassinosteroids, as well as their antagonistic interactions with ABA, positively regulate the seed germination process [8,13,14]. During seed germination, ABA acts as an inhibitor, promoting seed dormancy, which is vital for preventing premature germination under unfavorable conditions [15]. The balance between ABA and GA during seed germination determines the outcome of this process [16]. High concentrations of ABA inhibit germination, while GA promotes it. Under drought stress, ABA levels increase, activating ABA-responsive genes and enhancing plant tolerance to drought. ABA plays an essential role in both seed germination and resistance to environmental stresses, making it a key regulator of seed germination and early seedling growth [17,18].

ABI5 is a bZIP TF that plays a pivotal role in the ABA signaling pathway [19]. It is activated by SnRK2 kinases (SnRK2.2, SnRK2.3, and SnRK2.6), which phosphorylate the transactivation domain of ABI5 in vegetative tissues under stress conditions [20]. Phosphorylation alters the conformation of ABI5, enhancing its ability to interact with other proteins [21]. Once activated, ABI5 binds to the promoter regions of ABA-responsive genes, such as *RD29A*, *RD29B*, and *EM6*, and upregulates their expression [15]. Notably, *EM1*, *EM6*, and *LEA D-34* (encoding late embryogenesis abundant (LEA) proteins) were among the first identified direct targets of *ABI5* [22]. Yeast one-hybrid assays further confirmed that ABI5 directly binds to the ABRE motifs in the *EM6* promoter [21,23]. Several *ABI5* target genes, such as *PGIP1* and *PGIP2*, are involved in seed germination by inhibiting polygalacturonase activity, thereby delaying seed coat rupture and suppressing germination [24]. In this process, *ABI5* is negatively regulated by *PED3*, a peroxisomal ABC transporter involved in fatty acid β-oxidation.

Plant developmental processes such as dormancy and germination, as well as interactions with the environment, are regulated by phytohormones and other signaling molecules. Previous studies have demonstrated that plant hormones and other signaling compounds play crucial roles in regulating both developmental pathways and responses to abiotic stresses [5]. Glutathione (GSH), one of the major antioxidant molecules, is essential for plant survival under stress conditions by detoxifying excess reactive oxygen species (ROS), maintaining cellular redox homeostasis, and modulating protein function [25]. Recently, GSH has also emerged as an important signaling molecule that modulates ABA signaling and related developmental events, as well as stress responses. In Arabidopsis guard cells, GSH was shown to negatively regulate ABA signaling components, including excluding ROS production and Ca^2+^ oscillations, thereby suppressing ABA-induced stomatal closure [26]. Increased levels of GSH-related proteins/compounds, such as glutaredoxins (GRXs) and S-nitrosoglutathione (GSNO), have been shown to suppress ABA signaling during seed germination [27]. Moreover, glutathione transferases (GSTs) involved in the GSH metabolic pathway have been reported to affect the expression of *ABI5* [28], suggesting a complex interaction between GSH and ABA signaling, particularly during the critical phase of seed germination. Given that *ABI5* is a key gene in ABA-mediated seed dormancy and germination, it is, therefore, essential to investigate the potential regulatory relationship between *ABI5* and glutathione metabolism.

Rapeseed *(Brassica napus* L.) is an important oilseed crop, widely cultivated for edible oil and meal production. The drought tolerance of rapeseed is crucial for its agricultural performance. Although ABA and *ABI5* play important roles in drought tolerance, the specific functions of *BnABI5* (the homolog of *ABI5*) in seed germination and early seedling growth in rapeseed have not been fully elucidated. In this study, Genome-Wide Association Studies (GWAS) of drought phenotypes during seed germination and early seedling growth identified *BnABI5* as being associated with drought stress in rapeseed. To explore the role of *BnABI5* in drought tolerance during seed germination and early seedling growth, we generated *BnABI5* mutants using Clustered Regularly Interspaced Short Palindromic Repeats/CRISPR-associated protein 9 (CRISPR/Cas9) technology and compared their germination and early growth performance under normal and drought stress conditions with WT plants. To investigate the impact of the *BnABI5* mutation on seed germination and drought tolerance in rapeseed, we analyzed 60 RNA-seq samples collected from two genotypes, three treatments, and four time points. Differential expression analysis and Weighted Gene Co-Expression Network Analysis (WGCNA) were performed to uncover the regulatory networks involving *BnABI5* under drought conditions. These findings provide a foundation for the development of drought-tolerant germplasm in rapeseed breeding.

## 2. Materials and Methods

### 2.1. Plant Materials and Germination Rate Calculation Under Drought Treatment

In this study, 196 rapeseed accessions were obtained from the Crop Research Institute at Zhejiang University, China. All plant materials were planted in September 2022 at the Wuchang experimental base of the Oilseed Crop Research Institute, Chinese Academy of Agricultural Sciences (114.14° E, 30.32° N). Within each row, individual plants were spaced 0.15 m apart, with row spacing maintained at 0.25 m. Seeds were sown at the end of September and harvested in early May of the following year. Mature seeds were collected, sun-dried, and stored at −20 °C until further use. For each accession, 50 viable seeds were placed in petri dishes for germination, with each dish lined with two layers of filter paper (Shanghai Experiment Reagent Co., Shanghai, China). As a control, 7 mL of deionized water was used to moisten the filter paper, or a polyethylene glycol (PEG) 6000 (Solarbio, Beijing, China) solution was used to adjust the osmotic potential to −0.8 MPa to simulate drought stress conditions. The petri dishes were incubated in a growth chamber set (Jinggong Industrial Co., Shanghai, China) at a constant temperature of 25 °C, with light intensity of 100 μmol m^−2^ s^−1^, a 12-h light/12-h dark photoperiod, and 70% relative humidity. A completely randomized design was employed to ensure three replicates for each accession under both control and drought conditions. After seven days, the survival rate of each accession was recorded.

### 2.2. GWAS Analysis

We obtained a total of 2,404,340 SNPs from the rapeseed SNP database (BnaSNPDB: https://bnapus-zju.com/bnasnpdb/, accessed on 14 June 2023), filtering out SNPs with a minor allele frequency (MAF) less than 0.05 and a missing genotype rate greater than 0.5 to ensure high-quality data. GWAS analysis of drought survival rate phenotypes for each accession was performed using the Efficient Mixed-Model Association eXpedited (EMMAX) method [29] available on the BnaVGD platform (https://bnapus-zju.com/gwas/; accessed on 9 November 2023). This model accounts for population structure and relatedness, improving the accuracy of the association tests. Manhattan and quantile-quantile (Q-Q) plots were generated based on the GWAS results on the BnaVGD platform, with a significance threshold of −log10 (*p*-value) = 5.75 to minimize false-positive associations while capturing biologically meaningful signals.

### 2.3. CRISPR/Cas9-Mediated Editing of ABI5 in the B. napus

We performed gene editing of *ABI5* in *B. napus* following the method described by Li et al. [30], with the specific procedures as follows: We designed two sgRNAs with minimal off-target effects for the conserved sequences of three copies of *BnABI5* genes (*BnA04g0179250.1*, *BnA05g0194800.1*, and *BnC04g0676540.1*) using CRISPR-P 2.0: sgRNA 1 (TTTGGTCTGAGATACATAGAGG) and sgRNA 2 (GTATGGAGTTGATATGGGAGGG). The sgRNA assembly and vector construction followed protocols from previous studies [31,32].

Genome editing vectors were introduced into *Agrobacterium tumefaciens* strain GV3101 competent cells (Sangon Biotech, Shanghai, China) following the manufacturer’s protocol. The vectors containing the two sgRNAs were then introduced into the rapeseed variety “Zhongshuang 6” (ZS6) [33] using *Agrobacterium*-mediated transformation, with kanamycin as the selection marker. In the T0 generation, putative transgenic plants were first selected on kanamycin-containing medium, and PCR amplification using *NPTII* (a kanamycin resistance marker) primers (forward: 5′-GATGGATTGCACGCAGGT-3′; reverse: 5′-TCGTCAAGAAGGCGATAGA-3′) was performed to confirm the presence of the transgene and eliminate false positives. Verified T0 plants were transferred to a growth chamber (16 h light/8 h dark, 22 °C) for further growth and seed collection. T1 and T2 generation plants were grown under controlled conditions (22 °C, 16 h light/8 h dark) or at the Hanchuan Experimental Station (OCRI-CAAS). Genomic DNA was extracted from the leaves of these plants using the CTAB method [34]. For mutation detection, genomic regions spanning the CRISPR target sites were first amplified using gene-specific primers DT1-BsF (5′-ATATATGGTCTCGATTGtttggtctgagatacatagGTT-3′) and DT1-BsR (5′-ATTATTGGTCTCGAAACTCCCATATCAACTCCATACCAA-3′). The PCR products were then screened via PAGE gel electrophoresis to determine the presence of heterozygous or biallelic mutations based on banding patterns. Samples with distinct mutation bands were subsequently selected for Sanger sequencing. To ensure accurate mutation characterization, a nested PCR was performed using secondary primers DT1-F0 (5′-TGtttggtctgagatacatagGTTTTAGAGCTAGAAATAGC-3′) and DT1-R0 (5′-AACTCCCATATCAACTCCATACCAATCTCTTAGTCGACTCTAC-3′) to produce shorter amplicons (150–350 bp) for sequencing.

### 2.4. Stress Treatment Experiment for WT and BnABI5 CRISPR-Edited Lines

First, surface sterilization of WT (ZS6) and *BnABI5* CRISPR-edited line seeds was performed by soaking them in a 10% sodium hypochlorite (Ganzhou Jinfu Fine Chemical Technology Co., Ltd., Ganzhou, China) solution for 5 min. The seeds were then evenly placed on moist filter paper (Sunnyo Environmental Technology, Shanghai, China), with 50 seeds per germination box. Three treatment groups were set up: the control group (water treatment), the ABA treatment group (2 µM ABA solution), and the PEG treatment group (10% PEG solution). For each treatment group, the corresponding solution was added to maintain moisture on the filter paper. The treated seeds, along with the filter paper, were placed in 10 cm × 10 cm Petri dishes (Bickman Biotechnology, Changsha, China), with three replicates per group to ensure reliable results. Finally, the Petri dishes were placed in a plant growth chamber with a light/dark cycle of 16 h light/8 h dark and a light intensity of 300 µmol·m^−2^·s^−1^.

### 2.5. Phytohormone Quantification

The endogenous levels of indole-3-acetic acid (IAA), abscisic acid (ABA), jasmonic acid (JA), salicylic acid (SA), gibberellins (GA1, GA3, GA4, GA7), and 1-aminocyclopropane-1-carboxylic acid (ACC) were determined using liquid chromatography-tandem mass spectrometry (LC-MS/MS). Fresh plant tissues from WT and *BnABI5* CRISPR-edited lines at 0, 24, 48, and 72 h under ABA, PEG treatments, and controls were lyophilized, homogenized, and extracted with methanol/water/formic acid (15:4:1, *v*/*v*/*v*) containing deuterated internal standards (e.g., D2-IAA, D6-ABA). After centrifugation and purification via C18 solid-phase extraction, the supernatant was evaporated and reconstituted in methanol (Sigma-Aldrich, St. Louis, MO, USA). Chromatographic separation was performed on a C18 column with a gradient of methanol and 0.1% formic acid. Analytes were detected in multiple reaction monitoring (MRM) mode with electrospray ionization (ESI), using optimized precursor-to-product ion transitions for each compound (TSQ Fortis platform, Thermo Fisher Scientific, Waltham, MA, USA). Quantification was achieved by comparing peak areas to standard curves normalized against corresponding internal standards. Three biological replicates were analyzed per treatment.

### 2.6. Library Preparation and Transcriptome Sequencing

Total RNA was extracted from 60 samples (WT and *BnABI5* CRISPR-edited lines at 0, 24, 48, and 72 h under two treatments) using the RNAprep Pure Plant Kit (Tiangen Biotech, Beijing, China). RNA quality was assessed using a NanoDrop^®^ 2000 (Thermo Fisher Scientific, Waltham, MA, USA), and treated with DNase I (RNase-free) according to the manufacturer’s instructions. RNA purity and integrity were evaluated by agarose gel electrophoresis. RNA concentration was measured using the Qubit RNA BR Assay Kit (Q10210, Thermo Fisher Scientific, Waltham, MA, USA), and RNA integrity was checked using an Agilent Bioanalyzer 2100 system (Santa Clara, CA, USA). For RNA-seq library preparation, 5 μg of RNA from each sample was used. RNA libraries were constructed by Qingke Biotechnology Co. (Wuhan, China), using the Illumina TruSeq Stranded Library Prep Kit, and sequencing was performed on the Illumina HiSeq 2500 platform (San Diego, CA, USA). In brief, poly(A) mRNA was isolated from total RNA using Oligo-(dT) magnetic beads. RNA fragments were generated in fragmentation buffer, and the first-strand cDNA was synthesized using six-base random primers. Then, double-stranded cDNA (ds-cDNA) was synthesized using a buffer, dNTPs, RNase H, DNA polymerase I, and RNase H. Library fragments were purified using AMPure XP beads (Beckman Coulter, Brea, CA, USA) to obtain cDNA fragments of approximately 250–300 bp in length. To ensure library quality, PCR products were purified with AMPure XP beads and assessed on an Agilent Bioanalyzer 2100 system. The index-labeled samples were clustered using the cBot clustering system of the TruSeq PE Cluster Kit v4-cBot-HS (Illumina, San Diego, CA, USA). Finally, the libraries were sequenced on the Illumina HiSeq-PE150 platform (Illumina, San Diego, CA, USA).

### 2.7. Transcriptome Analysis

Raw RNA sequencing data were processed using fastp v0.23.4 [35] with default parameters to remove low-quality reads and adapters, generating high-quality clean reads. Clean reads were then aligned to the *B. napus* reference genome (Brana_ZS_V2.0, http://brassicadb.cn:82/download_genome/Brassica_Genome_data/Brana_ZS_V2.0/; accessed on 12 February 2024) using STAR v2.7.10b [36] with default settings. Gene read counts were obtained using featureCounts v2.0.3 [37], and gene expression levels were quantified in terms of Fragments Per Kilobase of exon model per Million mapped fragments (FPKM) using HTSeq 2.0 [38] with default parameters. Differentially expressed genes (DEGs) between samples were detected using DESeq v1.42.0 [39] in R v4.3.3, with the filtering criteria set to *P*_adj_ < 0.05 and |log_2_(FoldChange)| > 2. Cluster analysis of the 0, 24, 48, and 72-h time-point samples was performed using Mfuzz v2.60.0 [40]. Gene expression heatmaps were generated using the R package pheatmap v1.0.12 [41] (scale = “row”). GO and KEGG annotations of the protein sequences from Brana_ZS_V2.0 were performed using eggnog-mapper v2 [42] and kobas [43], respectively. Enrichment analysis was conducted with the R package clusterProfiler 4.0 [44], and results were visualized using ggplot2 v3.5.1 [45].

### 2.8. WGCNA

Gene co-expression network analysis was performed using the R package WGCNA v1.73 [36]. From the expression matrix of 50,488 genes, the top 25% of genes with the highest variance in expression across samples were selected as the input dataset. Hierarchical clustering of all samples was conducted using the hclust function with the average linkage method (method = “average”). The soft-thresholding power (β) was determined to be 16 using the pickSoftThreshold function. Co-expression modules were identified using the blockwiseModules function with TOMtype set to “unsigned”, minModuleSize set to 30, and mergeCutHeight set to 0.25. Module eigengenes (MEs) were calculated using the moduleEigengenes function, and the Pearson correlation coefficients (PCCs) between MEs and phenotypic traits such as ABA levels were computed with the *cor.test* function in R. Modules with a correlation coefficient greater than 0.65 and a *p*-value less than 0.05 were considered significantly associated with the phenotype.

### 2.9. Co-Expression Analysis of TFs and Genes

Protein sequences of TFs from all plant species were downloaded from the PlantTFDB 4.0 database [46]. These TF sequences were compared against the protein sequences of Brana_ZS_V2.0 using DIAMOND [47] with the blastp algorithm to identify TFs in the *B. napus* genome. The PCC between differentially expressed TFs and genes was calculated in R. Gene-TF pairs with |PCC| > 0.8 and Q-value < 0.05 were retained. Co-expression networks of these gene-TF pairs were visualized using Cytoscape v3.9.1 [48].

## 3. Results

### 3.1. BnABI5 as a Potential Regulator of Seed Germination Under Drought Stress in B. napus

In response to drought stress, *B. napus* seeds often regulate their germination through dormancy mechanisms, a strategy that helps the plant conserve energy and survive under adverse conditions. However, how *B. napus* seeds regulate their germination mechanisms under drought stress remains unclear. To identify potential factors influencing seed germination under drought stress in *B. napus*, we measured the germination rate of 196 *B. napus* accessions under simulated drought stress (Appendix A) and performed a GWAS using 2,404,340 SNPs obtained from the BnaSNPDB. The results identified two significant loci (C4:6775140 and C4:6775750) on chromosome C04, which are located inside the gene *BnaC04g09030D* (Darmor-bzh genome, *B. napus* v4.1, https://yanglab.hzau.edu.cn/BnIR/germplasm_info?id=Darmor.v4.1/, accessed on 9 November 2023) (Figure 1A). This gene is homologous to the Arabidopsis gene *AT2G36270* (*ABI5*), which is reported to play a crucial role in regulating plant development and responses to abiotic stresses [49]. Using homology searches, we identified three homologous genes in the ZS6 reference genome: *BnC04g0676540.1*, *BnA04g0179250.1*, and *BnA05g0194800.1*. Multiple sequence alignments revealed that these genes, like *AtABI5*, contain the typical bZIP domain (Figure 1B). These findings suggest that the *ABI5* homologs may play important roles in regulating seed dormancy and germination under drought stress in *B. napus*.

### 3.2. CRISPR Editing of BnABI5 Promotes Seed Germination Under Drought Stress Treatment

To further validate the function of *ABI5* homologs in *B. napus*, we designed sgRNAs targeting the three homologs of *ABI5* in the ZS6 reference genome and utilized the CRISPR-Cas9 gene-editing system for genome editing (Appendix A). The flanking sequences of the target sites were amplified and sequenced, confirming a 1 bp deletion at the sgRNA target site of *BnABI5* (Figure 2A, Appendix A). To evaluate the impact of drought stress treatment on seed germination in the edited lines, we treated WT and *BnABI5* CRISPR-edited lines with water, ABA, and PEG. After 24 h of treatment, the *BnABI5* CRISPR-edited lines displayed stronger growth compared to WT (Figure 2B). All seeds of both WT and the edited lines germinated after 48 and 72 h under different treatments. However, the *BnABI5* CRISPR-edited lines exhibited faster growth and longer shoots than WT under all treatments, with the most pronounced differences observed under water and PEG treatments (Figure 2C,D). Furthermore, we examined the germination rates of rapeseed seeds at multiple time points following treatment. The results indicated that the BnABI5 CRISPR-edited lines displayed an accelerated germination rate during the early stages under ABA and PEG treatments (Figure 2E).

### 3.3. Comparative Transcriptome Analysis of WT and BnABI5 CRISPR-Edited Lines Under Different Treatments

ABI5, as a TF, exerts systemic effects on the entire plant by regulating multiple downstream genes. To investigate the impact of *BnABI5* gene editing on plant responses to drought, this study performed transcriptomic sequencing on two genotypes of rapeseed under different treatments (H_2_O, ABA, PEG) at three time points (24 h, 48 h, and 72 h). Additionally, we compared the differential expression of genes in *BnABI5* CRISPR-edited lines relative to the WT under different treatments at these time points. With the exception of the 48 h PEG treatment, the number of downregulated DEGs in *BnABI5* CRISPR-edited lines was significantly higher than the upregulated DEGs at most time points (Figure 3A). This suggests that gene editing of *BnABI5* may disrupt some of the pathways in which *BnABI5* is involved. The enrichment analysis of these DEGs revealed that the downregulated genes in *BnABI5* CRISPR-edited lines were primarily enriched in GO terms associated with drought stress, such as S-glycoside metabolic process, defense response by cell wall thickening, and defense response by callose deposition in the cell wall (Figure 3B). Interestingly, in all treatments at different time points, the downregulated DEGs in *BnABI5* CRISPR-edited lines were consistently enriched in the glutathione metabolism pathway (Figure 3C). This indicates that *BnABI5* plays a crucial role in regulating cellular stress responses. Furthermore, after 24 h of PEG treatment, the upregulated genes in *BnABI5* CRISPR-edited lines were mainly enriched in pathways related to photosynthesis, glycolysis/gluconeogenesis, and carbon fixation in photosynthetic organisms (Appendix A). This may be due to the faster germination of seeds in *BnABI5* CRISPR-edited lines compared to WT, which are likely in a phase of rapid growth and development.

Additionally, we analyzed the overlap of DEGs between *BnABI5* CRISPR-edited lines and WT under different conditions. As the treatment time increased, the DEGs exhibited distinct overlap patterns, with varying numbers of overlapping DEGs between the *BnABI5* CRISPR-edited lines and WT under different treatment conditions. The PEG treatment at 72 h induced the most gene expression changes (Figure 3D), suggesting that *BnABI5* plays a significant role in regulating gene expression responses under prolonged drought stress. A total of 211 genes showed differential expression across all treatments. The cluster heatmap divided these genes into two groups: 184 genes that were highly expressed only in the WT, and 27 genes that were highly expressed in *BnABI5* CRISPR-edited lines (Figure 3E). These results provide further evidence that *BnABI5* gene editing alters the expression of key genes related to stress tolerance pathways, influencing the plant’s ability to respond to drought stress.

### 3.4. Complex Gene Expression Patterns in BnABI5 CRISPR-Edited Lines and WT Plants

To further analyze the temporal and treatment-specific response patterns of DEGs under stress treatments, we performed clustering analysis using mfuzz on 1179 DEGs that were differentially expressed in at least four out of six treatment groups. Based on the time-course dataset, these genes were assigned to 8 clusters, each containing between 33 and 286 genes (Figure 4A). The expression patterns of these clusters revealed that genes in cluster1, cluster4, cluster6, and cluster8 were predominantly highly expressed in WT plants, whereas genes in cluster2 and cluster7 were more highly expressed in *BnABI5* CRISPR-edited lines (Figure 4B).

To further explore the functions of genes within these clusters, we conducted enrichment analysis for the genes in each cluster. The results showed that genes in cluster1 and cluster4 were mainly involved in biological processes related to indole compound metabolism (such as indole-containing compound biosynthetic process, indolalkylamine metabolic process, and indole-containing compound metabolic process) and defense responses associated with callose deposition and cell wall thickening (such as defense response by cell wall thickening, defense response by callose deposition in cell wall). These processes are likely linked to plant defense mechanisms. In particular, the metabolism of indole compounds and the deposition of callose are important pathways through which *BnABI5* regulates plant stress responses. Additionally, genes in cluster8 were enriched in processes related to S-glycoside and glucosinolate metabolism, further suggesting that *BnABI5* may influence plant drought responses through the regulation of these metabolic pathways (Figure 4C). Furthermore, all DEGs in cluster7 showed higher expression levels in *BnABI5* CRISPR-edited lines compared to WT, and their expression increased over the course of seed germination. These genes were enriched in biological processes related to copper ion response (response to copper ion) and superoxide (superoxide), which may be related to the ability of plants to enhance stress tolerance under drought by regulating these pathways. The analysis of gene expression patterns and functions in these clusters provides strong evidence for understanding the effects of *BnABI5* gene editing on seed germination and the dynamic changes in gene expression during seed germination.

### 3.5. Gene Expression Regulatory Networks Involving Multiple Hormones

The above analysis indicates that the mutation of *ABI5* induces dysregulation of several genes, particularly under stress conditions, revealing a complex regulatory network between *ABI5* and other genes. This network likely involves interactions with various plant hormones, such as ABA, jasmonic acid (JA), and 1-aminocyclopropane-1-carboxylic acid (ACC), which are known to play critical roles in stress responses. To further explore this complex gene-hormone interaction, we conducted a WGCNA to identify gene modules with similar expression patterns across all samples (Figure 5A). These genes were grouped into 12 modules, with the largest module, MEturquoise, containing 3927 genes, and the smallest module, MEgreenyellow, containing only 44 genes (Figure 5B).

Since plant responses to stress are often initiated by rapid changes in hormone levels, we calculated the correlation between eight plant hormones: indole-3-acetic acid (IAA), JA, ABA, ACC, gibberellin 1 (GA1), gibberellin 2 (GA2), gibberellin 4 (GA4), and salicylic acid (SA), and the identified expression modules (Appendix A). Analysis of the relationship between the modules and hormonal traits revealed significant correlations, with MEgreenyellow being significantly associated with ABA, MEblack with JA, and MEpurple with ACC (*p* < 0.05) (Figure 5C). Thus, the MEgreenyellow module emerges as a key module containing genes related to ABA synthesis and ABA-regulated genes involved in seed growth and development in *B. napus*. We further performed functional enrichment analysis on the genes in the MEgreenyellow module. The results indicated that 44 genes were enriched in biological processes associated with stress responses, including indole-containing compound biosynthetic process (24%), glucosinolate metabolic process (19.6%), and defense response by callose deposition (10.7%) (Figure 5D). KEGG pathway enrichment showed significant enrichment in glutathione metabolism and phenylalanine, tyrosine, and tryptophan biosynthesis pathways. These findings suggest that genes in the MEgreenyellow module may influence ABA synthesis and drought resistance in *B. napus* by regulating the metabolism and biosynthesis of glutathione, phenylalanine, tyrosine, and tryptophan.

### 3.6. Regulation of Glutathione Metabolism Pathway Genes by TFs

Given the important role of GSH in the ABA signaling pathway, this study further investigates the co-expression relationship between genes enriched in the glutathione metabolism pathway and TFs identified through WGCNA. We first identified TFs in the *B. napus* genome, identifying a total of 7702 TFs. The results show that 9 genes in the glutathione metabolism pathway are co-expressed with 99 TFs, which belong to 32 different TF families. The MYB family contains the highest number of TFs (11), followed by C2H2 (9), Trihelix (7), NAC (7), bZIP (7), and WRKY (6) (Figure 6A, Appendix A).

Further analysis revealed that 34 TFs exhibit differential expression between *BnABI5* CRISPR-edited lines and WT, including 5 MYB, 3 WRKY, 3 HSF, 3 C2H2, and 3 bZIP TFs. Most of these differentially expressed TFs show the highest expression levels in the water treatment at 0 h, with expression levels decreasing in the samples after different treatments. Furthermore, the expression levels of these differentially expressed TFs in WT are generally higher than in *BnABI5* CRISPR-edited lines. This suggests that editing the *BnABI5* gene may affect the expression of these TFs, thereby regulating the gene expression in the glutathione metabolism pathway and influencing the ABA signaling and drought response mechanisms in plants. These findings provide new insights into the molecular mechanisms by which *BnABI5* regulates the glutathione metabolism pathway and may offer theoretical support for improving drought resistance in *B. napus*.

### 3.7. The Role of Glutathione-Mediated ABA Signaling in Drought Resistance

Drought-induced increases in GSH levels enhance the expression of ZEP (zeaxanthin epoxidase) and NCED (9-cis-epoxycarotenoid dioxygenase), promoting the conversion of β-carotene to abscisic aldehyde. Additionally, GST (glutathione S-transferase) mediates detoxification reactions and promotes the expression of *ABI3* and *ABI5*, thereby activating ABA-responsive genes and enhancing plant drought resistance [50] (Figure 7A). In the GSH-mediated drought regulatory pathway, 70 genes were differentially expressed in *BnABI5* CRISPR-edited lines compared to WT, including 39 *GST*, 13 *PP2C* (*protein phosphatase 2C*), 11 *PYR/PYL*, 3 *ABI5*, 3 *ABI3*, 1 *ZEP*, and 1 *NCED*. Notably, *BnC09g0896470.1* (*ZEP*) and *BnA05g0215860.1* (*NCED*) showed increased expression levels with prolonged ABA and PEG treatments (Figure 7B), and *ZEP* expression was higher in *BnABI5* CRISPR-edited lines than in WT. Two *ABI3* genes (*BnC03g0586980.1* and *BnA03g0136860.1*) and three *ABI5* genes (*BnA05g0194800.1*, *BnA04g0179250.1*, and *BnC04g0676540.1*) exhibited high expression only at 0 h under water treatment, with reduced expression after treatment. Among the 39 *GST* genes, genes such as *BnC08g0878140.1*, *BnA08g0329980.1*, and *BnA07g0302050.1* had higher expression levels in *BnABI5* CRISPR-edited lines than in WT under drought (PEG) treatment. For *PP2C*, *BnA07g0297030.1* showed elevated expression only in *BnABI5* CRISPR-edited lines after 72 h of PEG treatment, while the other 12 *PP2C* genes were generally expressed at low levels. Three *PYR/PYL/RCAR* genes (*BnA07g0285090.1*, *BnC03g0563340.1*, and *BnC06g0754380.1*) exhibited increased expression after stress treatments, with overall higher expression in *BnABI5* CRISPR-edited lines compared to WT. This suggests that GSH enhances ABA synthesis, thereby promoting the expression of *PYR/PYL/RCAR* while suppressing *PP2C*.

## 4. Discussion

This study identified *BnABI5* as a key candidate gene for seed germination under drought stress through GWAS. *ABI5* is a well-known functional gene that plays a crucial role in both seed germination and drought stress responses in plants [51]. By generating stable genetic lines of *BnABI5* CRISPR-edited plants, we observed an increased seed germination rate at 24 h under water, PEG, and ABA treatments. Although at 72 h the germination rates of the mutant plants were similar to those of the WT plants, the *BnABI5* CRISPR-edited lines exhibited stronger growth, suggesting that *BnABI5* knockout facilitates seed germination under drought conditions.

As a TF, the phenotypic changes caused by the mutation of BnABI5 are mediated through its involvement in a complex gene regulatory network. To investigate which genes are affected by the *BnABI5*-mediated regulatory network, this study designed time-course RNA-seq datasets under two genotypes and various treatment conditions to provide insights into the role of *BnABI5* in seed germination and drought stress. The analysis revealed that, under most treatment conditions and time-points, the number of downregulated DEGs in the mutant plants was significantly higher than the number of upregulated DEGs. Further analysis of the expression trends of DEGs between the *BnABI5* CRISPR-edited lines and WT plants highlighted the functions of these genes, especially those showing lower expression levels in the WT (DEGs in clusters 1, 4, 6, and 8). Notably, the genes in these clusters (except cluster 6) were enriched in the glucosinolate biosynthesis pathway, in which enzymes are involved in regulating ABA levels and the synthesis of other secondary metabolites [52,53]. After successful *ABI5* gene editing, the plants exhibited stronger root development at 72 h, possibly due to the disruption of ABA homeostasis resulting from the downregulation of certain enzymes in the glucosinolate biosynthesis pathway, which accelerated the seed germination process.

The results of this study demonstrate that *BnABI5* editing influences phenotypes through a complex regulatory network, affecting numerous genes across various pathways. For instance, it was observed that, at most stress treatment time points, downregulated genes in the mutant plants were enriched in the glutathione metabolism pathway. Moreover, genes in the MEgreenyellow module, identified through WGCNA as significantly associated with ABA levels, were also enriched in the glutathione metabolism pathway. GSH is an important signaling molecule that regulates ABA signaling and its role in seed dormancy and drought resistance [36]. Increased levels of phenylalanine, tyrosine, and tryptophan are linked to enhanced drought tolerance mechanisms in plants, including stomatal regulation, plant hormone synthesis, and oxidative stress responses [37,38]. Furthermore, ABA and GSH are closely related in mediating plant responses to drought stress [54,55,56]. GSH plays a role in regulating ABA-induced downstream signaling of H_2_O_2_ in guard cells, preventing stomatal closure. Additionally, GSH is involved in detoxifying H_2_O_2_ through the AsA-GSH pathway [57,58]. Through differential expression analysis and WGCNA, this study identified key genes in the glutathione metabolism pathway, including *BnC02g0518750.1* (*GST*), *BnA03g0120550.1* (*GST*), *BnA10g0413960.1* (*gshA*), and *BnA03g0147440.1* (*gshA*). These genes provide potential candidates for further investigation into how *ABI5* regulates downstream genes to achieve fine-tuned responses to drought stress. These results offer promising molecular markers for marker-assisted selection (MAS). For example, favorable alleles of *BnABI5* or its downstream targets could be used to develop drought-tolerant rapeseed cultivars through conventional breeding or gene editing approaches. Furthermore, the regulatory network revealed in this study provides a framework for selecting gene combinations that balance germination vigor and drought resilience, which could enhance seedling establishment in water-limited environments.

Several studies have shown that ABI5 can interact with other TFs to regulate target genes. For example, in apple, *ABI5* interacts with *MdbHLH3*, enhancing the expression of target genes in the anthocyanin biosynthesis pathway and thereby promoting anthocyanin production [59]. The WD40-repeat protein XIW1 in Arabidopsis has been shown to regulate ABI5 stability and contribute to the modulation of ABA signaling [60]. To explore how *ABI5* affects target genes through TF interactions, this study conducted a detailed analysis of the co-expression relationships between genes in the glutathione metabolism pathway and TFs. The results revealed that TFs co-expressed with genes in this pathway generally exhibited lower expression levels in *BnABI5* CRISPR-edited lines compared to WT plants (under most drought stress treatment conditions). Previous studies have shown that several TFs in the glutathione metabolism pathway negatively regulate *ABI5*. For instance, in Arabidopsis, *G6PD5* inhibits *ABI5*, thereby affecting ABA-mediated seed germination and root growth [61]. *CycC1;1* negatively regulates the ABA signaling pathway during seed germination by inhibiting *ABI5* [62]. Our results suggest that the expression of these TFs may be influenced by *ABI5* or its signaling pathway. The regulatory relationship between these TFs and *ABI5* warrants further experimental investigation. In summary, this study systematically elucidated how the candidate gene *ABI5* influences multiple genes and pathways through its regulatory network, thereby affecting germination and drought resistance phenotypes in rapeseed. This was achieved through an integrative approach, starting from germination rate assessments under drought treatment, candidate gene identification, mutant construction, and transcriptomic analysis. These findings lay a foundation and provide clues for identifying additional drought-resistance candidate genes. These findings lay a theoretical and molecular foundation for integrating ABI5-related genes into breeding programs aimed at improving drought tolerance in *B. napus*. Future studies could focus on validating these markers in diverse genetic backgrounds and testing their effectiveness under field conditions to accelerate their application in practical breeding strategies.

## Figures and Tables

**Figure 1 plants-14-01276-f001:**
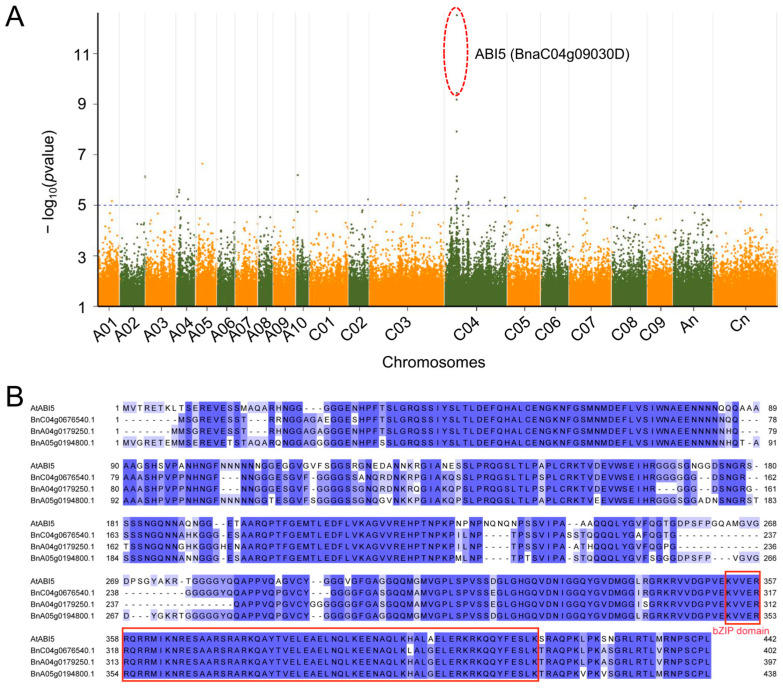
(**A**) The Manhattan plot shows the GWAS results for the drought survival rate phenotype. The dashed line represents the significance threshold, with a value of −log10 (*p*-value) = 5.75. The *x*-axis indicates the chromosomal positions, and the *y*-axis shows −log10 (*p*-value). The circled points indicate those SNPs located within *BnaC04g09030D*. (**B**) Multiple sequences alignment of putative amino encoded by *AtABI5* and the three *ABI5* homologous genes in the ZS6 reference genome. Identical amino acid residues across all four sequences are colored blue, with variants shown in light blue. The bZIP domain is indicated by a red box.

**Figure 2 plants-14-01276-f002:**
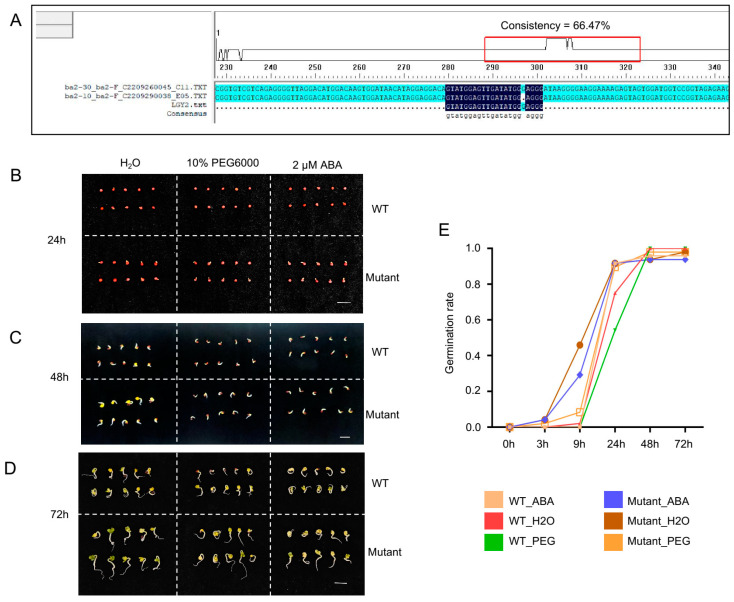
Gene editing of *ABI5* in rapeseed. (**A**) Sanger sequencing results of CRISPR-CAS9-edited *BnABI5*. In the alignment, blue highlights identical PCR product sequences, while the dark background indicates regions matching the sgRNA. (**B**–**D**) Phenotypes of WT and mutant (*BnABI5* CRISPR-edited lines) rapeseed seeds subjected to water, PEG, and ABA treatments at 24 h (**B**), 48 h (**C**), and 72 h (**D**). (**E**) Germination rates of rapeseed seeds under different treatments after different times. Mutant represents *BnABI5* CRISPR-edited lines.

**Figure 3 plants-14-01276-f003:**
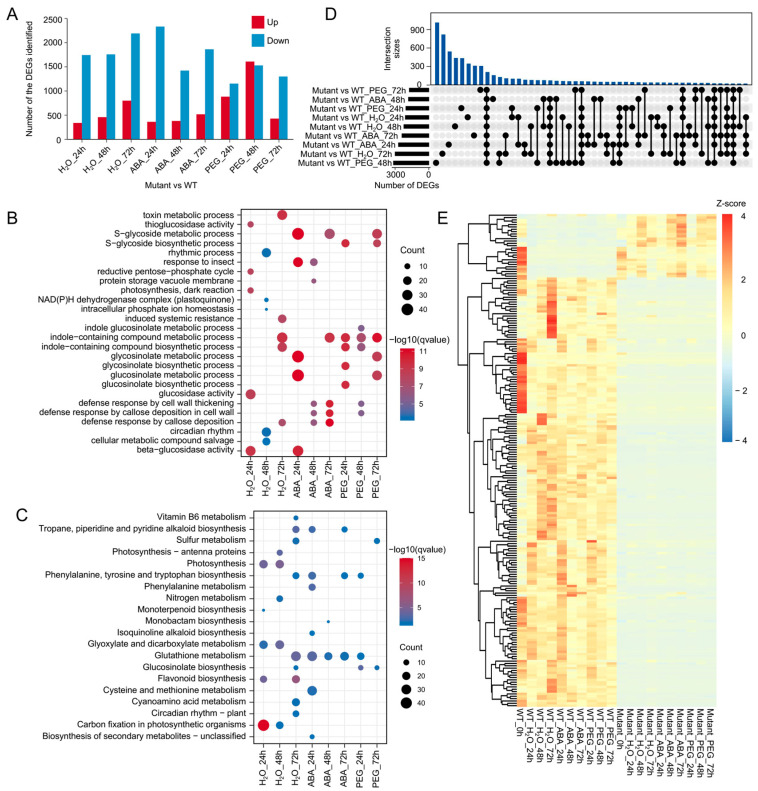
Differential expression analysis of *BnABI5* CRISPR-edited lines compared to WT under different treatments. (**A**) The number of upregulated and downregulated genes in *BnABI5* CRISPR-edited lines (mutant) under various treatments after different time points. (**B**,**C**) GO (**B**) and KEGG (**C**) enrichment analysis of downregulated genes in different treatment groups. (**D**) UpSet plot showing the number of unique and overlapping DEGs across 9 comparison groups. (**E**) Heatmap of gene expression for the 211 DEGs that are differentially expressed across all treatments. The color scale represents the Z-score of the gene expression level.

**Figure 4 plants-14-01276-f004:**
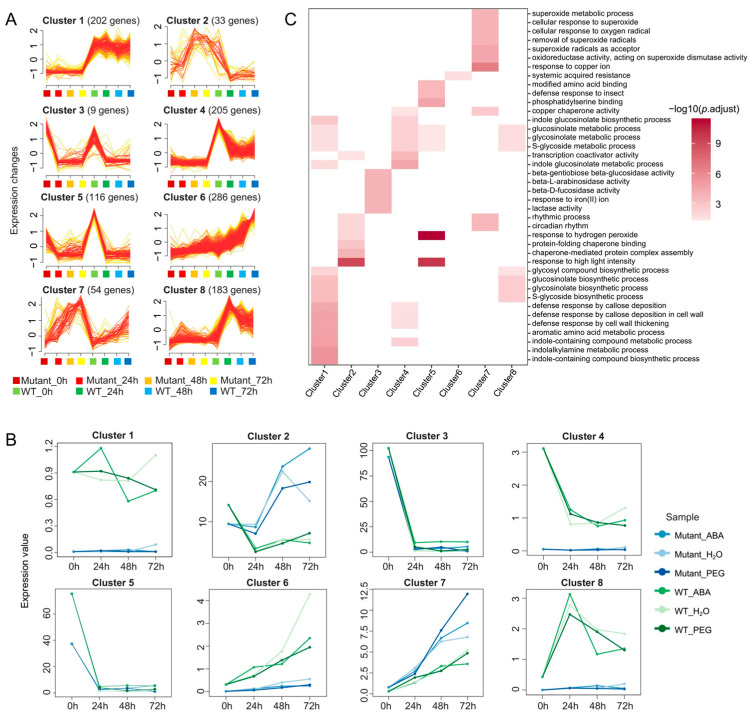
Mfuzz Clustering Analysis of DEGs. (**A**) Clustering of 1179 DEGs based on their expression patterns across four time points (0 h, 24 h, 48 h, and 72 h). The *x*-axis represents treatment duration, while the *y*-axis indicates expression changes. (**B**) Expression patterns of genes in the eight clusters under different treatments. The *x*-axis denotes treatment duration, while the *y*-axis shows the median expression level of all genes in each cluster. (**C**) Heatmap of the GO enrichment results of genes from eight distinct clusters.

**Figure 5 plants-14-01276-f005:**
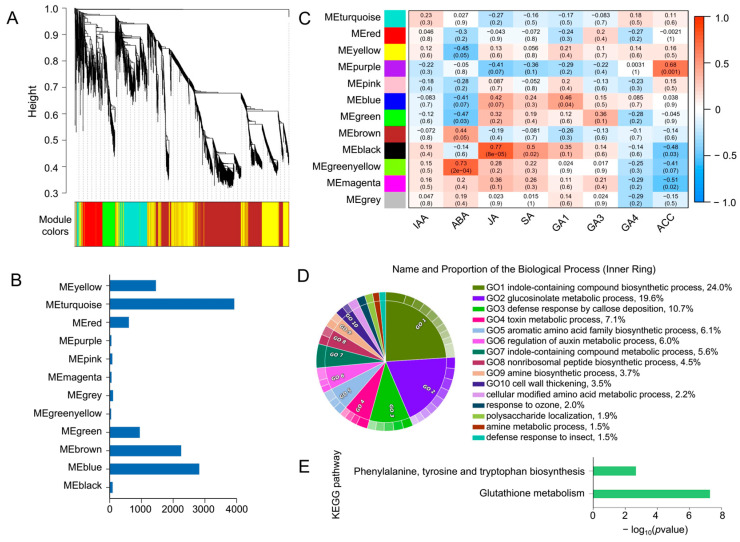
WGCNA of transcriptome and 8 hormone phenotypes. (**A**) Dendrogram displaying 12 co-expression modules identified by WGCNA across all samples. (**B**) Heatmap showing the correlation between the 12 modules and the levels of 9 plant hormones. Each row corresponds to a module represented by different colors, and each column represents one hormone. Red indicates a positive correlation between the module and the hormone, while blue indicates a negative correlation. (**C**) Bar chart illustrating the number of genes in each module. (**D**,**E**) GO (**D**) and KEGG (**E**) enrichment analysis results for the 44 genes in the MEgreenyellow module, which are significantly correlated with ABA.

**Figure 6 plants-14-01276-f006:**
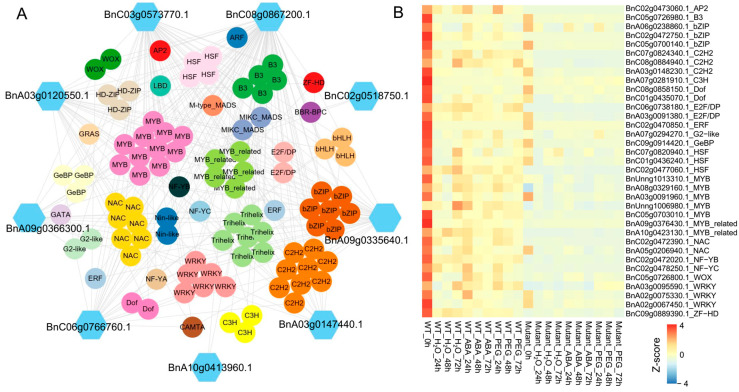
Co-expression of glutathione metabolism pathway genes and TFs. (**A**) Co-expression regulatory network between 9 genes in the glutathione metabolism pathway and TFs. (**B**) Heatmap of the expression of differentially expressed TFs in the co-expression network. The color scale represents the Z-score of the gene expression level.

**Figure 7 plants-14-01276-f007:**
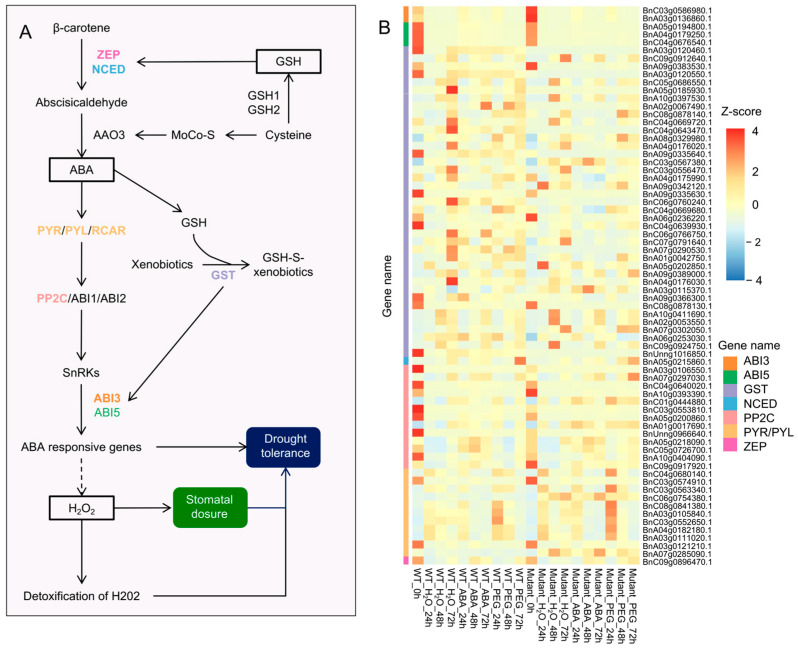
Glutathione-mediated ABA signaling regulates stomatal closure and drought resistance. (**A**) A pathway diagram illustrating the regulatory role of glutathione-mediated ABA signaling in stomatal closure and drought resistance. (**B**) Heatmap showing the expression profiles of 70 DEGs involved in the pathway. The color scale represents the Z-score of the gene expression level.

## Data Availability

The data are presented in the manuscript and the Appendix A. The raw read data are submitted to the Short Read Archive (SRA) and BioProject accession number PRJNA1227215.

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
