# Peer review of "ABA Enhances Drought Resistance During Rapeseed (Brassica napus L.) Seed Germination Through the Gene Regulatory Network Mediated by ABA Insensitive 5"

_plants, 2025, doi:10.3390/plants14091276_

Round 1

Reviewer 1 Report

Comments and Suggestions for Authors

The manuscript "ABA Enhances Drought Resistance During Rapeseed Seed Germination Through the Gene Regulatory Network Mediated by ABA Insensitive 5" describes a detailed study of the function of the BnABI5 gene during rapeseed germination. The BnABI5 gene is a key participant in the abscisic acid signaling pathway. The work is based on a comprehensive review and comparative analysis of wild-type plants and mutants of the BnABI5 gene obtained during the experiments. This work makes it possible to identify a whole range of drought-resistance candidate genes. The manuscript is presented consistently and logically and may be of interest to a wide range of readers. At the same time, there are a number of comments to the manuscript:

  1. The abstract must include the definition of the abbreviation DEG.
  2. 2. Lines 74-76 - the sentence content is more consistent with the discussion section, not the introduction. The introduction usually describes the relevance of the work, goals and objectives, a brief overview of previous research, the initial hypothesis, the relevance of the task.
  3. Why were only T2 plants tested for mutations? Why were they tested for the nptII gene first? During segregation in T1 and T2 generations, the transgenic construct and the target gene could have diverged into different plants. Why was mutation testing not performed immediately? Perhaps it is worth adding to the article that two guide RNAs should have led to the formation of a deletion, and it was this that was searched for using PCR. At what stage was kanamycin used as a selective marker if the plants were immediately grown in a growth chamber and not on a selective medium?
  4. To section 3.2. How was the presence of induced mutations in all three copies of the gene confirmed?
  5. If the ABI5 gene is one of the major TFs in the ABA signaling pathway, then the ABI5 mutants should have a different response to ABA, which is not observed in Figure 2E.
  6. Line 346 – which 60 samples are we talking about?
  7. There is little discussion with other works in the “Discussion”, and accordingly there are few references to other works. What does this manuscript contribute to our understanding of how the ABI5 gene works in plants? How does this information fit with data obtained in other plants? Have similar studies been conducted in other plant species and what did they show? What are the future prospects for this research?
  8. In the Results section, a number of sentences are more in line with the Discussion section; in particular, most of the sentences containing references to other works (for example, sentences from lines 363-367) could be moved (and expanded) to the Discussion.

Author Response

Thank you very much for taking the time to review this manuscript. The point-by-point responses to the reviewers' comments are provided in the attached file.

Reviewer 2 Report

Comments and Suggestions for Authors

I have read the article titled "ABA Enhances Drought Resistance During Rapeseed Seed Germination Through the Gene Regulatory Network Mediated by ABA Insensitive 5" with great interest. It explores the role of the transcription factor ABI5 in regulating drought tolerance during early seed germination in Brassica napus, using a combination of GWAS, CRISPR/Cas9 mutagenesis, transcriptomics, and network analysis. Overall, the manuscript is well written and easy to follow. However, there are several aspects of the manuscript that, if clarified, would increase the overall scientific rigor and practical relevance of the work. My specific comments and suggestions are provided below.
Introduction:
Lines 49–57. Consider briefly noting the crosstalk between ABA, JA, GA, and ethylene early in the introduction.
Lines 58–63. The description of ABI5 functions is too brief. Please mention known ABI5 targets and interactions in Brassica napus or model species to strengthen the justification for its functional study.
Line 68. The word Genome should not be written in lowercase at this point.
– A considerable portion of the results is devoted to the glutathione metabolism pathway. Please add a paragraph summarizing glutathione’s known role in ABA-mediated stress tolerance.
Materials and Methods:
Line 114. Provide a description of the heat shock method. Alternatively, you may include a reference to the original publication in which this method is described in detail.
Line 122. If a proprietary isolation kit was not used, a link or citation to the detailed method is also required.
Line 127. The method used to verify CRISPR/Cas9-induced mutations is described too briefly. Although a reference is provided, it does not explain the principle by which this method detects genome edits. Since this is a fundamental aspect of the study, the authors must provide a more detailed description of the method or cite a publication that clearly explains its application in detecting gene editing events.
Line 138. The light intensity of 300 μmol m⁻² s⁻¹ is quite high. Please clarify whether this value could be excessive or stressful for the seedlings, and if not, provide a reference supporting its use in similar experiments.
Line 201. Please indicate which software program was used to perform the statistical analyses.
Results:
Line 220. B. napus should be written in italics.
Figure 2E. It appears that the authors omitted error bars and statistical significance indicators for the observed differences. Please revise the figure to include these elements.
Discussion:
– The study has strong relevance to agriculture; however, this aspect is not fully explored in the discussion. Please expand the discussion to include the possible application of the results in marker-assisted selection or other strategies for integrating the findings into Brassica napus breeding programs.
– At the end of the discussion, please briefly mention potential directions for future research.

Author Response

(The authors gave the same response as above.)

Round 2

Reviewer 1 Report

Comments and Suggestions for Authors

All comments on this manuscript have been taken into account by the authors. The manuscript can be recommended for publication in the journal.